# Convective response to large-scale forcing in the Tropical Western Pacific simulated by spCAM5 and CanAM4.3

Toni Mitovski[1], Jason N. S. Cole[1], Norman A. McFarlane[2], Knut von Salzen[2], Guang J. Zhang[3]

[1]Canadian Centre for Climate Modelling and Analysis, Environment Canada, Toronto, ON, Canada
[2]Canadian Centre for Climate Modelling and Analysis, Environment Canada, Victoria, BC, Canada
[3]Scripps Institution of Oceanography, University of California, San Diego, CA, USA

*Correspondence to*: Jason N. S. Cole (jason.cole@canada.ca)

**Abstract.** Changes in the large-scale environment during convective precipitation events in the Tropical Western Pacific simulated by version 4.3 of the Canadian Atmospheric Model (CanAM4.3) is compared against those simulated by version 5.0 of the super parameterized Community Atmosphere Model *(*spCAM5). This is done by compositing sub-hourly output of convective rainfall, convective available potential energy (CAPE), CAPE generation due to large-scale forcing in the free troposphere (dCAPE$_{LSFT}$), and near surface vertical velocity ($\omega$) over the time period May-July 1997. Compared to spCAM5, CanAM4.3 tends to produce more frequent light convective precipitation ($< 0.2$ mm h$^{-1}$) and underestimates the frequency of extreme convective precipitation ($> 2$ mm h$^{-1}$). In spCAM5 5 % of convective precipitation events lasted less than 1.5 h and 75 % lasted between 1.5 and 3.0 h while in CanAM4.3 80 % of the events lasted less than 1.5 h. Convective precipitation in spCAM5 is found to be a function of dCAPE$_{LSFT}$ and the large-scale near surface $\omega$ with variations in $\omega$ slightly leading variations in convective precipitation. Convective precipitation in CanAM4.3 does not have the same dependency and instead is found to be a function of CAPE.

## 1 Introduction

Global climate models (GCMs) typically have a horizontal grid-scale that is much larger than individual deep convective clouds which requires parameterizations of convection and its effect on the large-scale atmosphere. Convective and other GCM parameterization often have adjustable parameters "tuned" within their range of uncertainty so that the model simulates reasonable climatological distributions of temperature, clouds, and wind fields (Mauritsen et al., 2012). However, these long-term climatological averages are the result of many shorter time-scale subgrid convective events which may have their biases masked by averaging. It is known that climate models tend to exhibit less rainfall variance than observations (Scinocca and McFarlane, 2004; DeMott et al., 2007), tend to produce light precipitation ($< 10$ mm day$^{-1}$) more often than observed (Zhang and Mu, 2005b; Sun et al., 2006; Dai, 2006) and underestimate the occurrence of extreme precipitation events (Wilcox and Donner, 2007; Boyle and Klein, 2010; Wang et al., 2017).

Studies have attributed biases in simulated precipitation variability to convective parameterizations employed in models (Zhang and Mu, 2005b; DeMott et al., 2007; Wang and Zhang, 2013; Wang et al., 2016). In general, convective parameterizations require a closure, which may or may not be activated (triggered) based on whether certain conditions are satisfied. If activated, the closure computes the cloud base mass flux which in turn is used to compute fluxes of mass, moisture, and precipitation above cloud base. Although convective precipitation is generated in response to the closure, it has been shown that the design of the trigger function may also influence the simulation of precipitation (Truong et al., 2009). Commonly used convective schemes employ triggers and closures based on net column moisture convergence

(Tiedtke, 1989) or convective available potential energy (Zhang and McFarlane, 1995) while some convective schemes use grid-scale upward motion in the lower troposphere as a trigger function (Donner, 1993; Bechtold et al., 2001).

A super-parameterization framework (Grabowski, 2001; Khairoutdinov and Randall, 2001) has been used to replace conventional convective and boundary layer parameterizations with cloud system resolving models (CSRMs) in GCMs.

When evaluated against observations and compared to GCMs, super-parameterized GCMs show improved tropical rainfall variability associated with the El Niño Southern Oscillation (Stan et al., 2010), the Madden-Julian Oscillation (Thayer-Calder and Randall, 2009; Kooperman et al., 2016), improved light and extreme precipitation over the US (Li et al., 2012), and more realistic diurnal cycle of summertime convection over mid-latitude continents (Guichard et al., 2004; Khairoutdinov et al., 2005). Therefore, super-parameterized GCMs may be useful to provide guidance for improving sub-

grid convective parameterizations. However, the additional computing cost for the super parameterization implementation is prohibitive for most modeling groups, which is a major reason why it is still not widely used.

In the analysis that follows we examine how models simulate deep tropical convective events, so it is worthwhile to summarize the behavior one might expect. Tropical convective clouds are often organized into a specific pattern known as the "building block model" (Mapes et al., 2006). Within this pattern, shallow convective clouds precede deep convective

clouds which are then followed by stratiform anvil clouds. Shallow convective clouds pre-moisten the lower-troposphere and thus support the growth of deep convective clouds (Johnson et al., 1999; Sherwood, 1999; Sobel et al., 2004), while deep convective clouds detrain large amounts of condensate in the upper-troposphere and therefore contribute to the development of stratiform anvil clouds. The stratiform clouds, with cloud base near the melting level (Zipser, 1977), generate about 40 % of the tropical precipitation (Schumacher and Houze, 2003). Falling through the unsaturated air under

the cloud base, some fraction of the stratiform precipitation evaporates, generating negatively buoyant downdrafts which may penetrate to the surface (Zipser, 1977). By mass continuity the stratiform downdrafts induce upward motion in the background atmosphere thus contributing to moistening and cooling of the lower-troposphere. The forced lift and the low-level moistening and cooling contribute to increasing low-level instability and thus may promote further initiation of new convection (Mapes, 1993; Mapes and Houze, 1995; Fovell et al., 2006). Some features of the building block models, the

shallow convective pre-moistening and the strength of the stratiform circulation, have not been realistically simulated in global climate models (Mitovski et al., 2010).

For this study we use sub-hourly output from Version 4.3 of the Canadian Atmospheric Model (CanAM4.3) and version 5 of the super-parameterized Community Atmospheric Model (spCAM5) to isolate strong convective precipitation events in each model for a 3-month period in the Tropical Western Pacific (TWP). To evaluate the ability of CanAM4.3 to simulate

convective precipitation relative to spCAM5 and the relationship between precipitation and the environment, composites of convective available potential energy (CAPE), CAPE generation in the free troposphere, large-scale near surface omega, and convective precipitation are analyzed for all convective events in this region.

## 2 Model description

Version 4.3 of the Canadian Atmospheric Global Climate Model (CanAM4.3) has several improvements relative to its
predecessor, CanAM4 (von Salzen et al., 2013), including improvements to parameterizations of radiation and land surface processes. CanAM4.3 uses a hybrid vertical coordinate system with 49 levels between the surface and 1 hPa, with a resolution of about 100 m near the surface. The triangular spectral truncation of the model dynamical core is T63, with an approximate horizontal resolution of 2.8 degrees latitude/longitude.

The mass flux scheme of Zhang and McFarlane (ZM) is used in CanAM4.3 to parameterize the effect of deep convection on the large-scale environment (Zhang and McFarlane, 1995). The diagnostic closure of Zhang and McFarlane (1995) in CanAM4 has been replaced with a prognostic closure (Scinocca and McFarlane, 2004). The diagnostic closure assumes that convection consumes CAPE at a rate that is proportional to the (positive) difference between the ambient value and some specified threshold value. The triggering condition is that CAPE is greater than zero. A quasi-equilibrium state could emerge if the large-scale CAPE production balances the convective consumption but it is not imposed a-priori. The prognostic closure also does not assume quasi-equilibrium a-priori but a quasi-equilibrium state could in principle emerge. The trigger condition in the prognostic closure is also CAPE greater than zero. When activated, the prognostic closure computes the cloud base mass flux which increases proportionally with CAPE and is then dissipated within a specified time scale.

To account for the effect of cumulus clouds with cloud tops below the ambient freezing level on the large-scale environment, CanAM4.3 employs a shallow convection scheme (von Salzen and McFarlane, 2002). The shallow convection scheme includes a parameterization of autoconversion processes to account for the effect of drizzle formation in shallow cumulus clouds following the approach in Lohmann and Roeckner (1996). The shallow convection scheme employs a diagnostic cloud base closure (Grant, 2001) based on a simplified turbulent kinetic energy budget for the convective boundary layer. The shallow scheme is not allowed to be active if the deep scheme is triggered at the same gridpoint and is vertically limited so that it operates mainly within the lower troposphere.

Version 5 of the Community Atmosphere Model (CAM5) used for the super-parameterized run has a horizontal resolution of 1.9° x 2.5° (latitude x longitude), 30 vertical levels from the surface to 3.6 hPa, and a time step of 1800s for the physical parameterizations (Neale et al., 2012). Version 5.0 of the super-parameterized Community Atmosphere Model (spCAM5) employs a 2D CSRM within each CAM5 grid cell to replace the convective parameterization of moist convection and other atmospheric parameterizations. The CSRM uses 32 columns each with 4 km horizontal grid-spacing and 28 vertical layers, between 992 and 14.3 hPa, coinciding with the lowest 28 levels in CAM5. Details of the CSRM and information on CSRM implementation within CAM can be found in Khairoutdinov and Randall (2001 and 2003) and Wang et al., 2011.

For both models the period of analysis is limited to the period between May 1st and July 24th of 1997 after each model simulation has spun up (1 January 1996 to 30 April 1997 for CanAM4.3 and 1 January 1997 to 30 April 1997 for spCAM5). For CanAM4.3, a six member ensemble was generated by uniquely adjusting the seed for the random number generator on 1 January 1997. This was done to improve the statistical representation of the results from this model as data from all ensemble members were used in the analysis. The spCAM5 spin up is done using CAM5. Output over the domain 150°E – 170°E and 0°N – 10°N is extracted and used for our analysis. Over this domain the 4-km 10-minute CSRM output from spCAM5 is used to compute the quantities needed for the analysis while output from CanAM4.3 is available every 15 minutes (the model dynamical timestep). Both models used monthly varying prescribed SSTs and sea ice fractions based on observations (Hurrell et al., 2008) as well as transient concentrations of trace gases and aerosols that are representative of conditions during the time period of the simulations.

Methodology

**3.1 Convective precipitation definition**

Within spCAM5, the CAM5 atmospheric parameterizations have been replaced by CSRMs, so it was necessary for our analysis to devise a method to separate the convective from the total precipitation. Convective precipitation in spCAM5 was

defined to be the total precipitation from all convective CSRM columns divided by the total number of columns (i.e. divided by 32). A CSRM column is categorized as convective if at any level the vertical velocity is greater than 1 m/s or less than -1 m/s and the sum of the cloud liquid and cloud ice water is greater than 0.1 g/kg. This definition is used in the studies of Suhas and Zhang (2015) and Song and Zhang (2018) which follows the study of Xu and Randall (2001) who set the thresholds based on examination of convective updraft and downdraft statistics in cloud-resolving model simulations of tropical and midlatitude convection. Convective precipitation in CanAM4.3 is generated within the deep and shallow convection schemes with the majority coming from the deep scheme.

The sensitivity of the results to the definition of convective precipitation from spCAM5 was evaluated by repeating the analyses using total instead of convective precipitation (Figs S1, S2, and S3). Results in Figure 1, 3(a), 3(c), and 4 were found to be similar using either the total or convective precipitation from spCAM5, implying insensitivity, for this study, to the exact definition of thresholds in the method of Suhas and Zhang (2015). This is mainly because in the regions being studied the precipitation was found to be mainly convective (> 70 %).

### 3.2 Convective event definition

Previous studies used observations (Mapes et al., 2006; Mitovski et al., 2010) and cloud-resolving model simulations (Suhas and Zhang, 2015) to isolate strong precipitation events and diagnose convection-environment interactions relative to the peak of these events. Although this is a useful diagnostic approach for the development of closure schemes, it lacks information regarding initiation of precipitation. For our analysis, we use a slightly different approach. An initiation time ($t_0$) of a convective event is defined as the time at which convective precipitation within a GCM grid box exceeds 0.01 mm h$^{-1}$ after following a 3-h period with no convective precipitation. An end time ($t_f$) of a convective event is defined as the time when convective precipitation, after exceeding 1 mm h$^{-1}$, drops down to less than 0.01 mm h$^{-1}$ within a time period of up to 12 h after initiation. Using this approach, we isolated 831 convective events in spCAM5 and 1452 in CanAM4.3 in the Tropical Western Pacific. Since the methodology isolates precipitating events that can last between 0.5 and 12 h, a scaled time (ST) is computed so that all events, regardless of lifetime, start and end at the same scaled time. The ST is calculated following Eq. (1):

$$ST(t) = \frac{(t - t_0)}{(t_f - t_0)} \times 100 \ \% \tag{1}$$

This approach improves comparison of composited events since features that precede or lag a rainfall peak, e.g. high CAPE and low convective inhibition (CIN) prior to peak rainfall and low CAPE and high CIN after peak rainfall, will occur at the same scaled time for all events regardless of lifetime.

### 3.3 Definition of convective available potential energy (CAPE) and CAPE generation

As defined in von Salzen and McFarlane (2002), CAPE, in J kg$^{-1}$, for an undiluted parcel of air rising from near the surface (SFC) to the level of neutral buoyancy (LNB) with the effect of condensate loading and without the effect of latent heat of fusion is calculated following Eq. (2):

$$CAPE = -g \int_{z_{SFC}}^{z_{LNB}} \frac{T_{vp} - T_{ve}}{T_{ve}} \, dz \tag{2}$$

where g is the gravity, $T_{vp}$ is the virtual temperature of a rising air parcel, and $T_{ve}$ is the virtual temperature of the large-scale environment.

CAPE as defined in Eq.2 includes two terms. The first term results from integration of the negative buoyancy between the surface and level of the free convection and represents the convective inhibition that the parcel of air has to overcome while it is lifted from the boundary layer into the convective layer. The second term results from integration over the region of positive buoyancy between the level of free convection and level of neutral buoyancy.

Following Zhang (2003, Eq. 5), CAPE is generated by radiative and advective large-scale processes $(\partial CAPE/\partial t)_{LS}$ and consumed by convective processes $(\partial CAPE/\partial t)_{CONV}$. The prognostic equation of CAPE is calculated following Eq. (3):

$$\frac{\partial CAPE}{\partial t} = \left(\frac{\partial CAPE}{\partial t}\right)_{LS} + \left(\frac{\partial CAPE}{\partial t}\right)_{CONV} \tag{3}$$

The large-scale (LS) generation term on the right hand side can be further separated into generation of CAPE by large-scale processes near the surface (dCAPE$_{LSS}$) and generation of CAPE by large-scale processes in the free troposphere (dCAPE$_{LSFT}$).

### 3.4 Definition of large-scale vertical velocity

By integrating the continuity equation in (x,y,p) coordinates, starting from the top of the atmosphere, ω (Pa s$^{-1}$) is computed from the mean divergence in a layer p using Eq. (4):

$$\omega_{p2} - \omega_{p1} = (p1 - p2)\left(\frac{\partial u}{\partial x} + \frac{\partial v}{\partial y}\right)_p \tag{4}$$

where omega at the top of the atmosphere is assumed to be zero, p1 is the pressure level above the layer p, and p2 is the pressure level under the layer p.

## 4. Results

### 4.1 Time-domain mean fields

Table 1 shows the domain (150°E – 170°E and 0°N – 10°N) and time (May – July) mean values for dCAPE$_{LSFT}$, ω, CAPE, and convective precipitation. Spatial standard deviations (in brackets) for each variable were computed using the time mean distributions over the domain. Both models show similar mean values for dCAPE$_{LSFT}$, ω, and convective precipitation. The CAPE values, however, are roughly 3-fold larger in spCAM5 (664 J kg$^{-1}$) than in CanAM4.3 (220 J kg$^{-1}$). For comparison, we computed CAPE using soundings from three tropical western Pacific sites: Truk/Caroline Is. (7.45° N and 151.8° E), Ponape/Caroline Is. (6.95° N and 158.2° E), and Majuro/Marshall Is. (7.08° N and 171.39° E). The observed May-July 1997 mean CAPE is 1080 J kg$^{-1}$.

The CAPE budget equation (Eq. 3) states that any change in CAPE between two time intervals is due to CAPE generation by the large-scale processes and due to CAPE consumption by convection during the two time intervals. It is known that in GCMs convection is activated too frequently (Zhang and Mu, 2005b) thus resulting in too frequent removal of CAPE and inability CAPE to accumulate to higher values. Since CanAM4.3 employs CAPE in its closure to compute mass flux and precipitation, too often activation will likely affect the precipitation rates resulting in too frequent too light precipitation. It

has been shown that GCMs tend to generate too frequent light precipitation (Sun et al., 2006; Dai, 2006; Wang et al. 2016) and underestimate the frequency of extreme precipitation (Wilcox and Donner, 2007; Boyle and Klein, 2010 ).

## 4.2 Frequency density of convective precipitation

Relative to spCAM5, CanAM4.3 overestimates the frequency of light convective precipitation (< 0.2 mm/h) and underestimates the frequency of extreme convective precipitation (> 2 mm h$^{-1}$) (Fig. 1b). Frequency density was defined as the ratio of the number of time steps with convective precipitation per 0.1 mm h$^{-1}$ convective precipitation bin to the total number of time steps. When compared to observations, models also exhibit less rainfall variance (Sun et al., 2006; Dai, 2006; DeMott et al., 2007; Mitovski et al., 2010).

We show that dCAPE$_{LSFT}$ (Fig. 1a) increases with convective precipitation intensity in spCAM5 and in CanAM4.3. In addition, omega ($\omega$) systematically increases with convective precipitation intensity in spCAM5 but not in CanAM4.3. For convective precipitation rates between 0.5 and 2.5 mm h$^{-1}$ CanAM4.3 shows a linear increase in $\omega$. Convective events, as defined in Sect. 3.2, can last between 0.5 and 12 h. Figure 1c shows the fraction of convective events, from the total number of convective events, as a function of the event length. Figure 1d shows the average peak convective precipitation as a function of the event length. About 5 % of the 831 spCAM5 events last less than 1.5 h and 75 % of the events last between 1.5 and 3.0 h and only 1 % of the events last longer than 5 h (Fig. 1c) with the most intense convective precipitation being associated with longer lasting events (Fig, 1d). In comparison, 80 % of the 1452 events in CanAM4.3 are shorter than 1.5 h and only 0.1 % of the events last longer than 5 h with the most intense convective precipitation being associated with shorter lasting events.

To examine sensitivity to the definition of convective precipitation (Sec. 3.1) we repeated our analysis using total instead of convective precipitation (Fig. S1). The results are similar except for the length of convective events and peak precipitation (Fig. 1d and Fig. S1d). This is due mainly to CanAM4.3 frequently producing light non-convective precipitation. This affected the event length while peak rainfall was more influenced by convective precipitation. This suggests that non-convective precipitation is more important for long lasting events.

While observations of sub-hourly precipitation are not available in the region used in this study, 3-hour observations of total precipitation is available from TRMM (Kummerow et al, 1998). To compare with the 3-hour TRMM data, 3B42v7 (TRMM, 2011), the spCAM5 and CanAM4.3 precipitation rates were averaged to 3-hour means. For the time and region used in this study, 3-hour mean total and convective precipitation are available from simulations using the Community Atmospheric Model version 5.1, CAM5.1, (Neale et al., 2012). Data from CAM5.1 is included since it uses a resolved physics that are similar to spCAM5 but like CanAM4.3 uses paramaterizations to represent subgrid-scale processes. The frequency distribution for total precipitation (Fig. 2b) shows that spCAM5 is more similar to TRMM than CanAM4.3 and CAM5.1 both of which are in turn more similar to each other than spCAM5 or TRMM. The similarity between CanAM4.3 and CAM5.1 and their difference relative to spCAM5 also holds for convective precipitation (Fig. 2a). These results suggest, at least for 3-hour means, the exact definition of convective rainfall is less important than differences between CanAM4.3 and spCAM5.

### 4.3 Relation between convective precipitation, large scale ω, and dCAPE$_{LSFT}$

In comparison to Figure 1a, which shows one quantity (dCAPE$_{LSFT}$ or ω) as a function of convective rainfall, Figure 3 shows convective rainfall histograms as function of two quantities. We find that convective precipitation in spCAM5 correlates best with both near surface -ω and dCAPE$_{LSFT}$ (Fig. 3a) with no dependence on CAPE (Fig.3c). Reversible and undiluted CAPE computed from semi-daily radiosonde profiles of temperature and humidity also shows that tropical precipitation intensity is not correlated with CAPE intensity (Mitovski and Folkins, 2014). As in Figure 1a, the strongest convective precipitation is associated with strong large-scale near surface ascent and strong dCAPE$_{LSFT}$. In addition, for a constant ω the rainfall rates increase with increasing dCAPE$_{LSFT}$ while for a constant value of dCAPE$_{LSFT}$ rainfall rates increase with decreasing ω (increasing ascent) with rain rates becoming more dependent on ω for larger dCAPE$_{LSFT}$. In the case when one of the quantities is in its lowest 25 percentile, for instance dCAPE$_{LSFT}$ < 50 J kg$^{-1}$ h$^{-1}$ or ω > 80 Pa s$^{-1}$, precipitation rates do not exceed 1 mm h$^{-1}$.

Precipitation simulated by CanAM4.3 (Fig. 3b) does not correlate with ω but does correlate with both CAPE and dCAPE$_{LSFT}$ (Fig. 3d) with greater rainfall rates being associated with larger values of CAPE and dCAPE$_{LSFT}$. This is expected since the precipitation generated within the ZM convection scheme is proportional to the updraft mass flux and the cloud water content. The updraft mass flux is closely related to the cloud base mass flux, which is computed within the prognostic CAPE based closure (Scinocca and McFarlane, 2004).

Repeating Fig. 3, using dCAPE$_{LSFT}$ and omega at levels ranging from 992 hPa to 232 hPa we found that, greater rainfall rates in spCAM5 is associated with more negative (stronger ascent) omega at 992 hPa and less negative omega at 232 hPa (not shown). Since omega was computed using Eq.4, a negative omega at pressure 992 hPa is approximately equal to the net column mass divergence above that pressure level.. Therefore, greater rainfall rates in spCAM5 are associated with strong low-level ascent or strong net column mass divergence and larger dCAPE$_{LSFT}$.

### 4.4 Composites over convective events

Prior the start of the convective event (time=0) in spCAM5 the near surface environment is characterized by a weak large-scale subsidence (Fig. 4a) and increasing relative humidity in the lower troposphere (Fig. 4g). An observed low-level moistening prior to deep convection has been previously attributed to moistening by shallow convective clouds (Sherwood, 1999; Sobel et al., 2004, DeMott et al., 2007). The moistening impacts the growth of convective clouds by modifying the dilution effect of entrainment on the buoyancy of rising air parcels (Sherwood, 1999; Raymond, 2000). The strength and depth of the pre-moistening are thus crucial in the transition from shallow to deep convection. The large-scale subsidence gradually weakens and diminishes about 20 min prior to time=0, roughly when CAPE reaches maximum and CIN reaches minimum (Fig. 4c). Therefore, a transition from a large-scale subsidence to large-scale ascent may be important in triggering convection. A near-surface omega tendency has been previously used as a trigger in the Donner convection scheme (Donner 1993; Wilcox and Donner 2007) in a version of the Geophysical Fluid Dynamic Laboratory (GFDL) Atmospheric model, version 3 (AM3) GCM. In their model, convection is triggered when near-surface omega becomes positive and exceeds a specified value and convective inhibition is less than 100 J kg-1. Although, dCAPE$_{LSFT}$ is positive prior to time=0 (Fig. 4e), precipitation is not initiated until ω becomes negative (large-scale ascent). The strongest ascent occurs around ST=45 %, shortly before the time of the strongest convective rainfall at ST=55 % and strongest dCAPE$_{LSFT}$ at

ST=65 %. Although dCAPE$_{LSFT}$ shows great similarity with the convective precipitation, it lags the precipitation by ST=5-10 % which may imply that large-scale generation of dCAPE$_{LSFT}$ during the event life-time may be a consequence of the model dynamics, i.e. response of the model to convective heating. During the decaying phase, after ST=75 %, dCAPE$_{LSFT}$ is still relatively strong but ω becomes positive (subsidence), CAPE reaches minimum, and CIN reaches maximum, which

likely prevent any further convection. Reversible and undiluted CAPE computed from 12-hourly radiosonde profiles of temperature and humidity shows similar behavior, with CAPE reaching a maximum prior to peak rainfall and minimum after peak rainfall (Mitovski and Folkins, 2014). The minimum CAPE and maximum CIN after peak rainfall are likely due to a combination of two effects, the export of boundary layer air with high moist static energy (MSE) into the middle troposphere by convective plumes and the injection of middle troposphere air with low MSE into the boundary layer by

mesoscale downdrafts (Zipser, 1977; Sherwood and Wahrlich, 1999). The effect of these two processes will also contribute to low-level drying, which is seen in spCAM5, the low- to mid-level dip in the relative humidity patterns that occurs after peak rainfall, but not in CanAM4.3.

In addition to examining the time evolution of dCAPE$_{LSFT}$ over convective events we compared its tendency (time change) and convective precipitation in spCAM5 (Fig. S4). We found that the tendency in dCAPE$_{LSFT}$ (Fig. S4) becomes positive

about 20 minutes prior to the start of a convective event (*t*=0) reaching a maximum slightly prior the precipitation maximum. While a thorough examine of this is beyond the scope of this paper we hypothesize that the trend in dCAPE$_{LSFT}$ could be associated with the trend in near surface omega. I.e., an increasing trend in the large-scale ascent contributes to increasing CAPE and positive trend in large-scale CAPE generation.

The large-scale environment prior the start of convective events in CanAM4.3 is quite different from spCAM5 with strong

ascent and relatively weak CAPE. CanAM4.3 shows some moistening prior to time=0, but this moistening occurs in a very shallow layer near the surface leaving the troposphere between 900 and 600 hPa relatively dry. The shallow mass flux patterns (not included) indicate that shallow convection is only active in the lowest 100 hPa. Relative to observations, GCMs also tend to have drier lower troposphere which has been linked to convection schemes employed in the model (Wang and Zhang, 2013). Convective rainfall is found to occur once CAPE exceeds 300 J/kg (Fig. 4d).  In contrast to

spCAM5, peak convective rainfall in CanAM4.3 (Fig. 4b) occurs closer to the end of the convective events corresponding with a peak in CAPE and CAPE generation (Fig. 4f). The strong correlation between convective precipitation and CAPE in CanAM4.3 is expected since convective precipitation in CanAM4.3 is proportional to cloud base mass flux which is in turn computed within the ZM prognostic closure as a function of CAPE (Scinocca and McFarlane, 2004). Convective precipitation in CanAM4.3 does not seem to correlate well with CIN (Fig. 4b) and this is likely because CIN is not

independently included in the ZM closure in CanAM4.3. Therefore, any discussion of CIN and linkage to CanAM4.3 precipitation is out of the scope of this paper. We should point out thought that, CIN is tightly coupled with precipitation over mid-latitude summertime continent but not with precipitation over oceans (Myoung and Nielsen-Gammon, 2010). The performance of various trigger functions and closures have been previously evaluated and it was found that in the tropics the best performing trigger functions are based on dCAPE$_{LSFT}$  and grid-scale vertical velocity in the lower

troposphere (Suhas and Zhang, 2014; Song and Zhang, 2017). Replacing CAPE with dCAPE$_{LSFT}$ in the ZM closure resulted in the National Center for Atmospheric Research Community Climate Model, version 3 (NCAR CCM3), simulating a more realistic Madden-Julian Oscillation (Zhang and Mu, 2005a), improved summer and winter mean tropical precipitation and less frequent light precipitation (Zhang and Mu, 2005b). Including a relative humidity at the parcel origin in the trigger function also improves the simulation of convection (Zhang and Mu, 2005b; Suhas and Zhang, 2014).

In general, most commonly used deep convection schemes in climate models employ closures based on CAPE or based on net column moisture convergence. We show that convective precipitation generated within a CAPE based closure is correlated to CAPE and CAPE generation in the free troposphere, while in spCAM5 precipitation is correlated to $dCAPE_{LSFT}$ and $\omega$. Since we computed $\omega$ from the horizontal winds starting from the top of the atmosphere (Eq. 4), near surface $\omega$ is closely linked with the net column mass convergence. Thus, it would be beneficial to compare correlation of convective precipitation generated within a net column moisture convergence based closure.

## 5. Summary

In the absence of high spatial resolution and sub-hourly observations, sub-hourly output from a super-parameterized AGCM (spCAM5) was used to study interactions between convective precipitation and the large-scale environment in the tropical western Pacific and to evaluate these interactions in a traditional AGCM (CanAM4.3). This is done by compositing model output of CAPE, CAPE generation in the free troposphere ($dCAPE_{LSFT}$), and large-scale near surface vertical velocity ($\omega$) over convective events during 1 May and 24 July 1997.

Although the domain mean convective precipitation, $dCAPE_{LSFT}$, and $\omega$ are found to be similar in the simulation period of May – July 1997 (Table 1), notable differences between CanAM4.3 and spCAM5 are found when compositing over convective events. The lengths of the convective events are shorter in CanAM4.3 with 80 % of the events lasting less than 1.5 h compared to 5 % in spCAM5. The strongest convective precipitation in CanAM4.3 is generated within shorter events while the strongest convective precipitation in spCAM5 is associated with longer lasting events. Compared to spCAM5, CanAM4.3 overestimates the frequency of light convective precipitation ($< 0.2$ mm h$^{-1}$) and underestimates the frequency of extreme convective precipitation ($> 2$ mm h$^{-1}$). When evaluated against observations, GCMs also tend to produce too frequent too light precipitation (Sun et al., 2006; Dai, 2006; Wang et al. 2016) which has been related to too frequent activation of CAPE based convective scheme (Zhang and Mu, 2005b).

Convective precipitation generated within spCAM5 is found to depend on both CAPE generation rate and near surface vertical velocity, two fields commonly used in trigger and closure functions of convective parameterization schemes. In CanAM4.3, which is representative of a CAPE-based closure scheme, convective precipitation is found to be a function of CAPE only (or CAPE generation). Interaction with the large-scale environment is found to differ between the models. In spCAM5, the maximum relative humidity is in the boundary layer roughly 1 h prior to t=0 (Fig. 4g). Increasing boundary layer moistening prior to peak rainfall seen in observations has been attributed to pre-moistening by shallow convective clouds prior to deep convection (Johnson et al., 1999; Sherwood, 1999; Sobel et al., 2004). The large-scale subsidence changes to large-scale ascent prior to t=0, and is coincident with the maximum CAPE value. After the initiation time in spCAM5, there is similarity between variations in $\omega$, $dCAPE_{LSFT}$, and convective precipitation, with variations in omega slightly preceding and variations in $dCAPE_{LSFT}$ slightly lagging variations in convective precipitation. In CanAM4.3 no dependence on $\omega$ was found, instead the model shows a dependence on CAPE and $dCAPE_{LSFT}$. The spCAM5 relative humidity patterns show a "dip" after peak rainfall, which has been previously linked to the injection of low moist static energy air from the middle into lower troposphere by mesoscale downdrafts (Zipser, 1977; Sherwood and Wahrlich, 1999). Although the relative humidity in CanAM4.3 has maximum in the boundary layer, this maximum is more persistent about peak rainfall and it occurs in a thin layer close to the surface. The height and time of the maximum humidity is coincident with the height and time of the shallow convective mass flux, suggesting that shallow convection, although important in moistening the boundary layer, does not penetrate to higher levels leaving the troposphere above 900 hPa relatively dry.

Although the sub-grid moist convection is a very complicated topic, in this study we see evidence that precipitation variability can be influenced by the design and the nature of the trigger and closure functions. The diagnostics described in this paper provide information regarding initiation and evolution of rainfall and can be used to study trigger conditions necessary for initiation of deep convection and the deep convection closure in regional and global models. We thus suggest that it is worthwhile to investigate the sensitivity of the precipitation generated within the ZM scheme in CanAM4.3 to various trigger and closure assumptions. For example, our results suggest that near-surface omega might provide a better trigger in combination with CAPE generation in the closure scheme used within CanAM4.3.

Code and data availability: Codes to perform the analysis described in the manuscript are available at https://github.com/jc-cccma/sub-hourly-convection-analysis with version 4.0.0 having the DOI:10.5281/zenodo.2619590.   Model output from spCAM5 and CanAM4.3 that are to be used as input for the codes can be found at https://zenodo.org/record/2658842 with the DOI:10.5281/zenodo.2658842.  Data from TRMM can be found at http://dx.doi.org/10.5067/TRMM/TMPA/3H/7. Model output from CAM5.1 was access from https://www.earthsystemgrid.org/dataset/ucar.cgd.ccsm4.cam5.1.amip.2d.001.atm.hist.3hourly_ave.html, accessed 20 March 2019.

Author contributions: TM conceived the method, performed the analysis and wrote the manuscript, JC designed and performed the CanAM4.3 simulations and contributed to writing the manuscript.  KS and NM provided expert advice to improve the analysis and the manuscript.  GJZ supplied spCAM5 model output and provided advice to improve the analysis.

Acknowledgments: The authors are grateful to Dr. Paul Vaillancourt and Dr. Rashed Mahmood for reviewing the manuscript. The authors also thank Chengzhu (Jill) Zhang of Lawrence Livermore National Laboratory for providing the spCAM5 output used in this study.

GJZ is supported by the U.S. Department of Energy, Office of Science, Biological and Environmental Research Program (BER), under Award Number DE-SC0016504.

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

Table 1: Domain (150°E – 170°E and 0°N – 10°N) and time (May – July 1997) means for spCAM5 and CanAM4.3 along with spatial standard deviations (in brackets) that were computed using the time mean distributions over the domain.

| Variable | SpCAM5 | CanAM4.3 |
|---|---|---|
| | Mean (St. Dev.) | Mean (St. Dev.) |
| $dCAPE_{LSFT}$ (J/kg/h) | 52 (19) | 50 (13) |
| $\omega$ (Pa/h) | -16 (6) | -8 (10) |
| CAPE (J/kg) | 664 (94) | 220 (67) |
| Convective prec. (mm/h) | 0.28 (0.10) | 0.28 (0.06) |

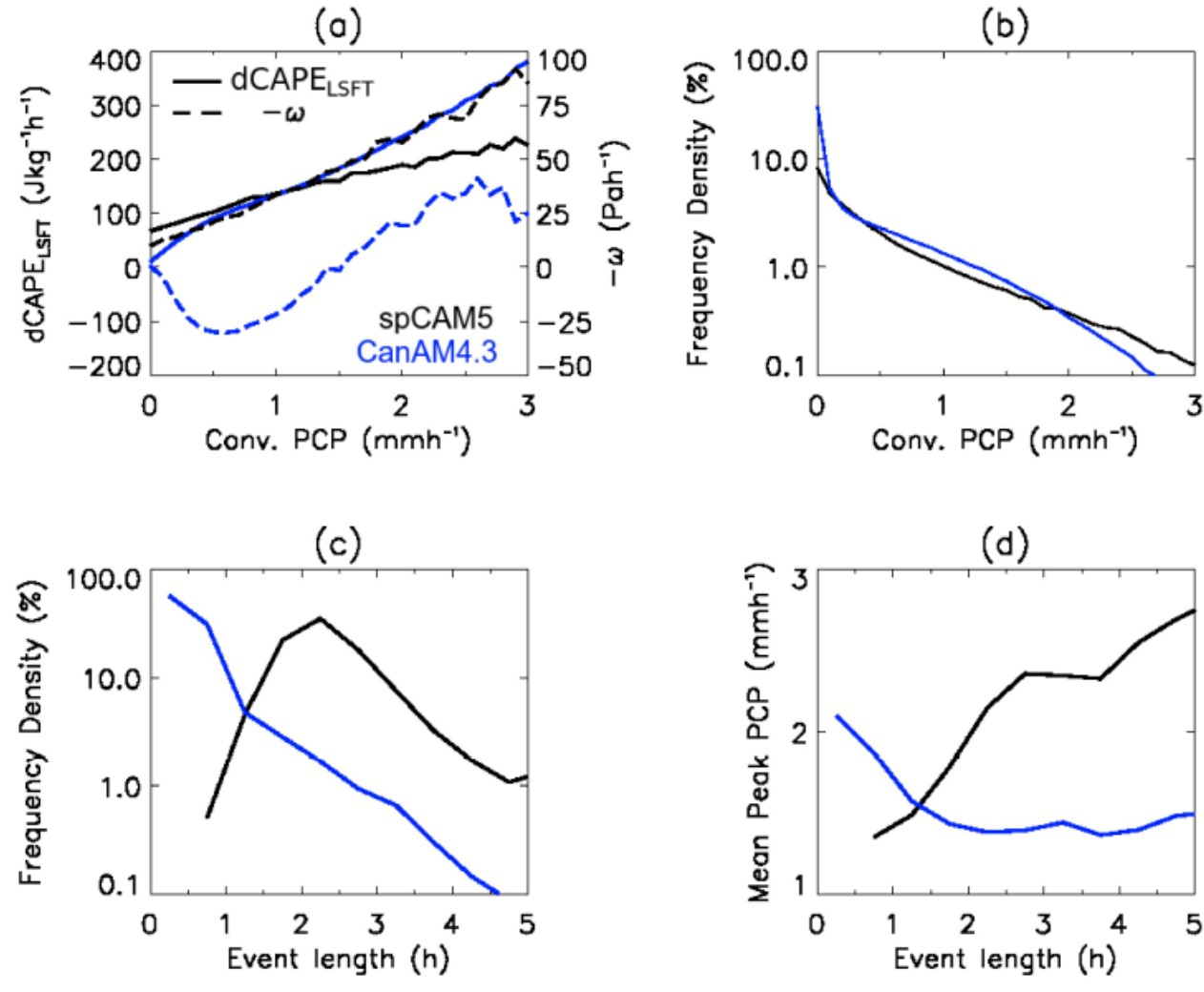

Figure 1: (a) Mean values of CAPE generation in the free troposphere (solid) and near surface large-scale -ω (dashed) per 0.1 mm/h convective precipitation bin in spCAM5 (black) and CanAM4.3 (blue), (b) frequency density of convective precipitation per 0.1 mm/h bin. Frequency density of 1 % in (b), or 1 mm/h convective precipitation in (a) and (b), corresponds to 82 CanAM4.3 and 122 spCAM5 samples per grid-cell. (c) frequency density of convective event length, (d) mean peak convective precipitation as function of convective event length. Frequency density of 10 % in (c) corresponds to 33 CanAM4.3 and 83 spCAM5 convective events.

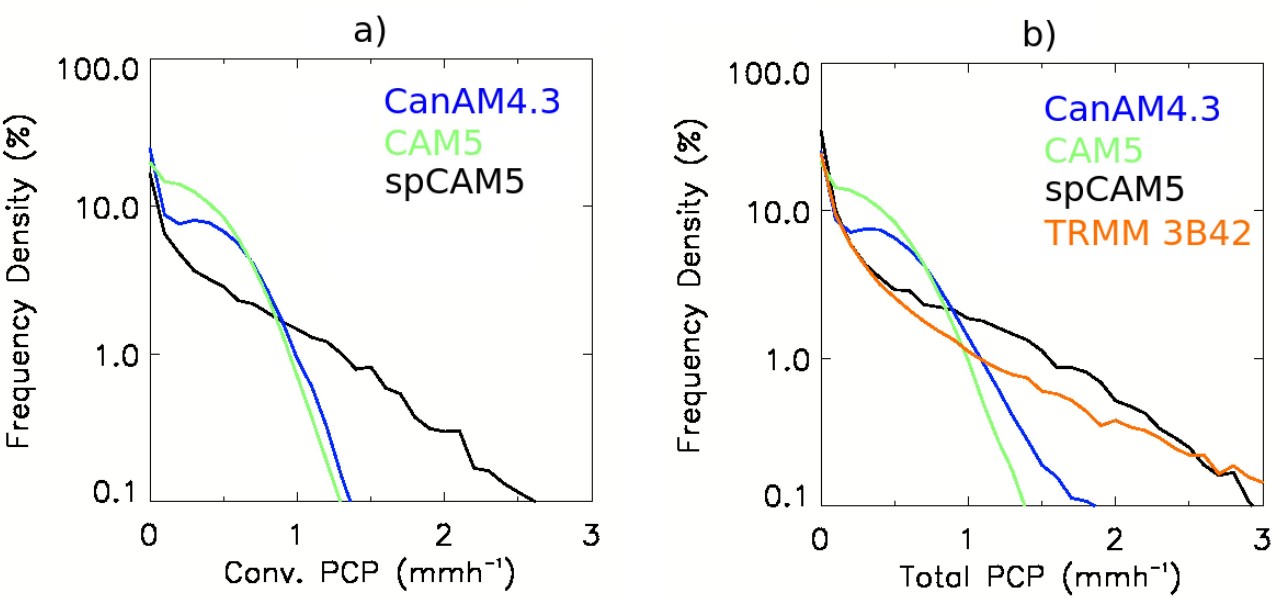

Figure 2: (a) Frequency density of 3-hourly convective precipitation per 0.1 mm/h bin, (b) frequency density of 3-hourly total precipitation per 0.1 mm/h bin.

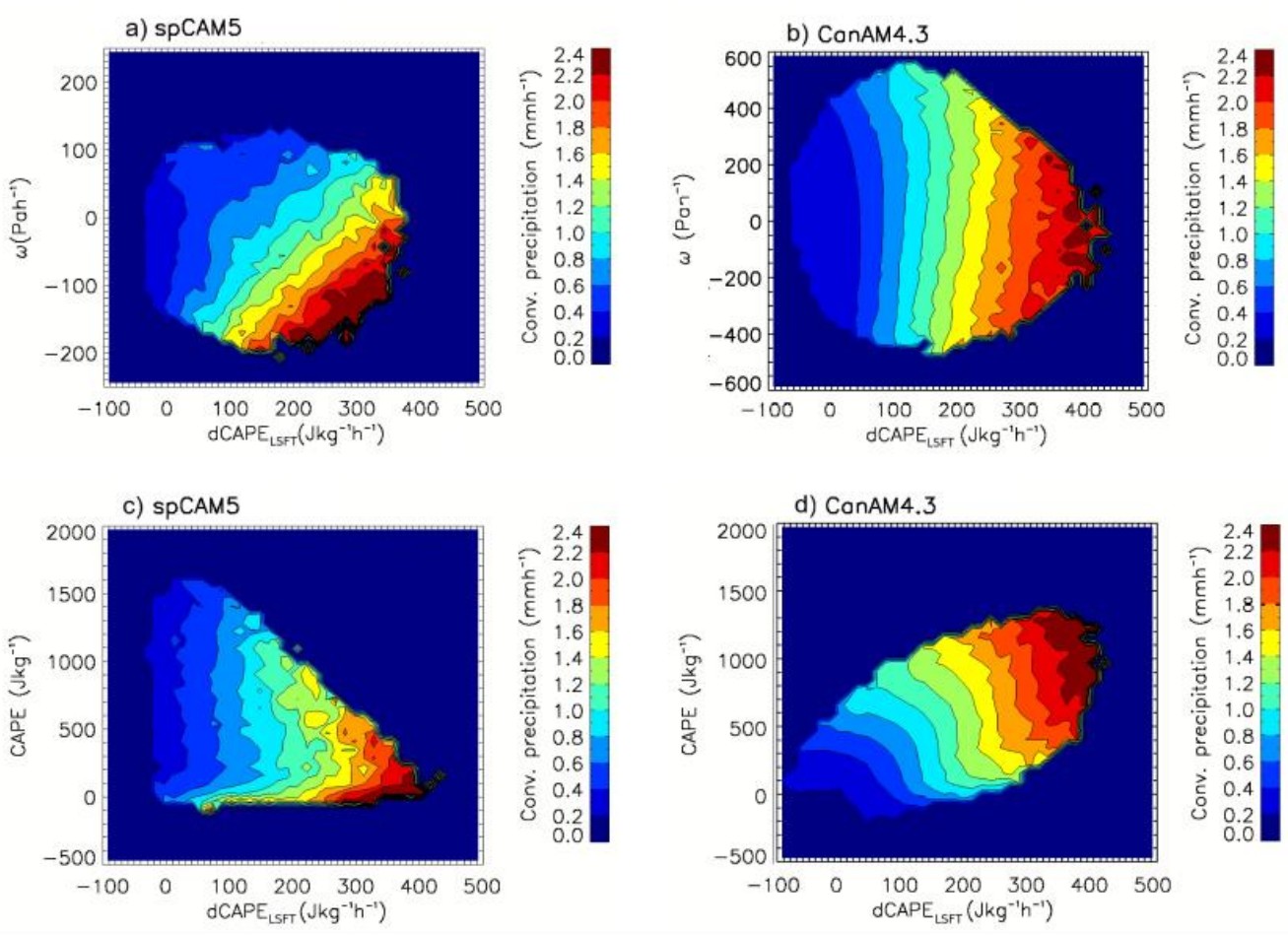

**Figure 3**: (a) spCAM5 and (b) CanAM4.3 mean convective precipitation as function of near surface ω and dCAPE$_{LSFT}$. (c) spCAM5 and (d) CanAM4.3 mean convective precipitation as function of CAPE and dCAPE$_{LSFT}$. Each of the plots consists of 1600 bins, 40 on X-axis and 40 on Y-axis. The convective precipitation within each bin is an average of at least 20 values.

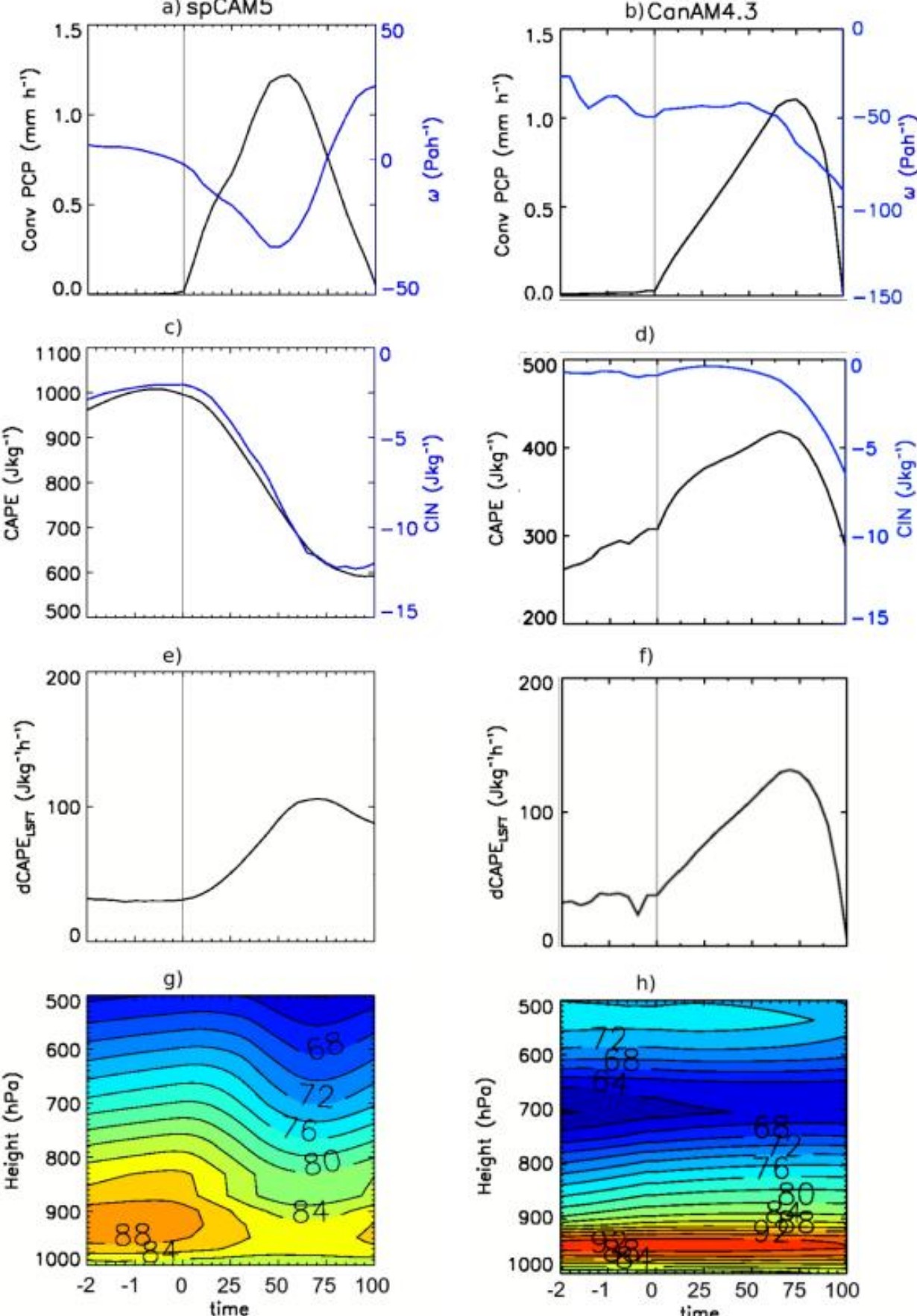

**Figure 4**: Prior to time=0, the time on the x-axis is in hours. After time=0, the time is in % of the event duration time. The rain events start at time=0 and end at t=100. The y-axis on the left shows values for the black curves and the y-axis on the right shows values for the blue curves. The panels on the left are for spCAM5 and the panels on the right are for CanAM4.3. The first row shows convective precipitation and large-scale near surface $\omega$; the second row shows CAPE and CIN; the third row shows dCAPE$_{LSFT}$, and the fourth row show relative humidity patterns.