# Peer review of "Convective response to large-scale forcing in the Tropical Western Pacific simulated by spCAM5 and CanAM4.3"

_Geoscientific Model Development, 2018_

## Referee Comment (RC1) · Anonymous Referee #1 · 6 Nov 2018

The paper is well written and the logic is easy to follow. They try to understand what the convection behaviors and its connection to large-scale environment in super-parameterized global model and global climate model. They found the behaviors are quite different in these two systems, with the super-parameterized model closer to the observation. It will provide useful information to design a good convection scheme for global climate model, and will help us to better understand the convection in the observation. I have a few major comments. 1. Why do you set the maximum lasting hour as 12 hour? It seems to me more reasonable if you don't set this one but only set the precipitation rate since there may be some convection events lasting longer than 12 hours. Have you checked that in the simulation, how much of the convection events

lasting longer than 12 hours? 2. Here, you checked the near-surface vertical velocity when considering the relationship between convection and large-scale environment. As shown in Song and Zhang (2017) you cited in the paper, the dCAPELSFT is mainly contributed by the vertical velocity and the vertical structure of large-scale vertical velocity is important for the convection development. Hence, could you also check the vertical structure of vertical velocity here? For example, similar to figure 2, could you also show the convective precipitation as function of different vertical velocity? In addition, in figure 2a, when dCAPELSFT is smaller than 50 J kg-1 h-1, convection precipitation is almost independent of near-surface vertical velocity, but when dCAPELSFT becomes larger and larger, the dependence of convective precipitation on the near-surface vertical velocity seems much tighter. It is a quite interesting phenomenon, maybe you can dig it further and check whether it is also the case for different levels of vertical velocity. 3. From Fig. 3, it seems that even for the dCAPELSFT, it is also not a good trigger for convection, since before and after convection (t=0), it doesn't change much (Fig. 3e). How can you set a threshold of dCAPELSFT to judge when the convection occurs. It is quite difficult. Instead, it seems that when convection happens, the tendency of dCAPELSFT becomes positive (d(dCAPELSFT)/dt). Have you further check the relationship between the convective precipitation and d(dCAPELSFT)/dt? 4. As shown in Song and Zhang (2018), the dCAPE-type triggers are significantly scale-dependent. In the higher-resolution models, it doesn't work very well compared to the coarser model resolution, since the relationship between dCAPE and convective precipitation becomes worse when the resolution is increased. Here, the spCAM5 is 4 km and CanAM4 is about 300 km. From figure 2, it seems that the relationship between convective precipitation and dCAPELSFT is much closer in CanAM4. Could you calculate the correlation and make some discussion about this issue. Reference: Song, F. and G. Zhang, 2018: Full Access Understanding and Improving the Scale Dependence of Trigger Functions for Convective Parameterization Using Cloud-Resolving Model Data, Journal of Climate, 7385-7399. 5. Finally, in the spCAM5, dCAPELSFT cannot be regarded as pure large-scale forcing, since it is calculated based on 4km

dataset (also see the discussion in Song and Zhang 2018). So how the convection happens in this model should be investigated further, since it provide more accurate description of convection. That will provide more information to the community.

---

## Referee Comment (RC2) · Anonymous Referee #2 · 30 Nov 2018

Overview:

In this study, the authors compared the representation of convective precipitation in the Canadian Atmospheric Model v4.3 (CanAM4.3) against the Superparameterized Community Atmosphere Model v5.0 (spCAM5). They evaluated composite fields of precipitation, convective available potential energy, and vertical velocity over the Tropical Western Pacific in simulations with prescribed SSTs for May-July 1997. The authors found that CanAM4.3 has more frequent light convective precipitation, less frequent extreme convective precipitation, shorter convective events, and convection that is less dependent on large-scale forcing compared to spCAM5. This paper is well written and

the results will be interesting to the modeling community. However, there are several areas that require additional analysis and more details in order to make this work more relevant and impactful. Major and minor comments are listed below.

Major Comments:

1. I encourage the authors to provide a deeper discussion and interpretation of the results. For example, the lack of relationship between convective precipitation and near surface vertical velocity (Figure 2b, 3b) and the mismatch in timing with CAPE/CIN in CanAM4.3 relative to spCAM5 are particularly interesting findings. However, to some degree, this has already been known and applied to improving the parameterization in the NCAR Community Atmosphere Model, as cited by the authors (Zhang and Mu, 2005a). How are the deficiencies in the parameterizations used in CanAM4.3 (i.e., CAPE based closure), which have been identified here, different from what is already known and published? And how can new information from the results presented here be applied to further improve models beyond what has already been implemented?

2. The effort to calculate "convective precipitation in spCAM5" in a way that is comparable to "convective precipitation in a parameterized model" is a great idea and potentially very useful. However, it is not clear that the way it is calculated in spCAM5 here means the same thing as it does from parameterized convection in CanAM4.3. How sensitive are the results to the values of the criteria (vertical velocity and cloud water/ice)? More importantly, how well does a definition of "convective precipitation " based on CRM vertical velocity and cloud water/ice match what "convective precipitation" means in a global parameterized model? Since the comparison and analysis is contingent on this calculation, it would be useful to discuss other ways it could be defined within spCAM5 and/or expected differences with what convective precipitation means in CanAM4.3. It would also be helpful to use an independent calculation of "convective precipitation" that could be applied identically to both models, which would likely be dependent on large-scale conditions. Ultimately, to what degree do the results and comparison between the models depend on the way that convective precipitation
has been defined? Likewise, how is CAPE calculated in spCAM5, is it at the CSRM or GCM scale? A comparison to CAPE calculated at the GCM scale would be most consistent with CAPE from CanAM4.3. Along these same lines, the differences in the relationship of convective precipitation and omega between spCAM5 (strong correlation) and CanAM4.3 (no correlation) may be, in part, due to differences in the definition of convective precipitation. I suggest including some analysis of relationships with "total precipitation rates" or alternative definition of "convective precipitation" in spCAM5.

3. In general, an explicit inclusion of observations for comparison would be helpful to the reader. The authors note that there is no dependence of convective precipitation with CAPE in spCAM5, which they say is consistent with observations by citing Mitovski and Folkins [2014]. It would be useful to make this calculation and include the observations in the figure for both CAPE and dCAPE. Likewise, the authors note that spCAM5's relationship between min/max CAPE and the timing of rainfall is consistent with observations by referring to Mitovski and Folkins [2014], but again I think showing the actual observations (as referenced) on the same figure would help.

Minor Comments:

1. Why not evaluate the ZM scheme as implemented in the conventional CAM5 to have more consistency with spCAM5? Many other aspects of the model are different between CanAM4.3 and spCAM5, beyond just the representation of convection, which makes the comparison somewhat unconstrained. I suggest including results from CAM5 as well as CanAM4.3 and spCAM5. Since only 3 months of simulation time is being assessed here and the initial setup of CAM5 would be the same as spCAM5, this should not add a significant amount of work.

2. I am confused about the vertical resolution used in spCAM5. Typically, the vertical resolution is 30 levels in the global grid and 28 levels in the CSRM (coinciding with the lowest 28 levels). Here the authors state that there are 66 levels CAM5, which would imply 38 levels above the CSRM rather than the typical 2 levels. Have pre-

vious studies used this configuration? Have you evaluated the differences between using 30 and 66 levels? Additionally, the Khairoutdinov and Randall (2001 and 2003) references are fairly old and refer to the implementation of super-parameterization in older versions of CAM. I recommend the authors cite more recent papers describing the implementation in CAM5, such as Wang et al., 2011 (https://www.geosci-model-dev.net/4/137/2011/gmd-4-137-2011.pdf).

3. Since spCAM5 is used instead of spCAM4, it includes aerosol processes and two-moment microphysics, so it might be helpful to describe these components of the model (MAM3 aerosol and Morrison microphysics) and compare them with the same processes in CanAM4. The representation of aerosol and cloud microphysics are likely to influence precipitation as well.

4. For the relationship between vertical velocity and convective precipitation in CanAM4.3 (Figure 1a), the authors conclude that "the results are not considered robust due to the few samples". Why not use more years for the CanAM4.3 results? CanAM4.3 is relatively cheap to run, so it is unnecessary for the authors to limit their analysis to such a short period. I recommend using more data, at least for CanAM4.3, to produce more robust results.

---

## Author Comment (AC1) · 16 Jan 2019

Thank you for your constructive comments, which were useful to improve our paper. Please see our responses below. The comments are in **bold** *italics* while the responses are in normal type.

***Why do you set the maximum lasting hour as 12 hour? It seems to me more reasonable if you don't set this one but only set the precipitation rate since there may be some convection events lasting longer than 12 hours. Have you checked that in the simulation, how much of the convection events lasting longer than 12 hours?***

We found that by setting the maximum lasting hour to 12 hours, we captured almost all events in CanAM4.3 and spCAM5. For instance, Figure 1(c) shows that less than 0.1 % of the events in CanAM4.3 last longer than 4.5 hours and less than 1 % of spCAM5 events last longer than 5 hours. Therefore, reducing the threshold to 6 hours or increasing the threshold to 24 hours will not affect the results in this paper.

The following text have been added to Section 4.2 in the manuscript:
"and only 1 % of the events last longer than 5 h"
"and only 0.1 % of the events last longer than 5 h"

***Here, you checked the near-surface vertical velocity when considering the relationship between convection and large-scale environment. As shown in Song and Zhang (2017) you cited in the paper, the dCAPELSFT is mainly contributed by the vertical velocity and the vertical structure of large-scale vertical velocity is important for the convection development. Hence, could you also check the vertical structure of vertical velocity here? For example, similar to figure 2, could you also show the convective precipitation as function of different vertical velocity?***

Additional Figure 2 (below) shows the convective precipitation as function of dCAPELSFT and vertical velocity at various levels, 232 hPa (top panel), 524 hPa, 763 hPa, 887 hPa, and 992 hPa (bottom panel). spCAM5 is on the left and CanAM4.3 is on the right. Additional Figure 2 shows that convective precipitation in CanAM4.3 has no dependency on omega but convective precipitation in spCAM5 it does depend on omega. In general, heavier precipitation in spCAM5 is linked to more negative (upward advection) omega at 992 hPa and less negative omega at 232 hPa. Since omega was computed from the large-scale horizontal winds starting from the top of the troposphere using the continuity equation, a negative omega at pressure p0 is approximately equal to the net column mass convergence above the level p0. Therefore, high rainfall rates in spCAM5 are associated with strong low-level ascent (net column mass convergence) and larger dCAPELSFT.

***In addition Figure 2(a) shows that, when dCAPELSFT is smaller than 50 J kg-1 h-1, convection precipitation is almost independent of near-surface vertical velocity, but when dCAPELSFT becomes larger and larger, the dependence of convective precipitation on the near surface vertical velocity seems much tighter. It is a quite interesting phenomenon, maybe you can dig it further and check whether it is also the case for different levels of vertical velocity.***

Indeed this is an interesting phenomenon. The left side panels in Additional Figure 2 (below) show that when dCAPELSFT is less than 50 J kg-1 h-1 convective precipitation

is nearly independent of near-surface omega, as well as omega at other levels. In addition, Figure 2(a) in the manuscript shows that, when dCAPELSFT is less than 50 J kg-1 h-1, precipitation varies between 0 and 1 mm h-1. But also, when the near-surface omega in Figure 2(a) is greater than 80 Pa s-1 (strong subsidence), convective precipitation is also nearly independent of dCAPELSFT and varies between 0 and 1 mm h-1. We can argue that, when one of the variables is in its lowest 25 percentile, convective precipitation in spCAM5 does not exceed 1 mm h-1.

The following text have been added to Section 4.3 in the manuscript:
"In the case when one of the quantities is in its lowest 25 percentile, for instance $dCAPE_{LSFT} < 50$ J kg$^{-1}$ h$^{-1}$ or $\omega > 80$ Pa s$^{-1}$, precipitation rates do not exceed 1 mm h$^{-1}$".

***From Fig. 3, it seems that even for the dCAPELSFT, it is also not a good trigger for convection, since before and after convection (t=0), it doesn't change much (Fig. 3e). How can you set a threshold of dCAPELSFT to judge when the convection occurs. It is quite difficult. Instead, it seems that when convection happens, the tendency of dCAPELSFT becomes positive (d(dCAPELSFT)/dt). Have you further check the relationship between the convective precipitation and d(dCAPELSFT)/dt?***

We thank you for this very useful suggestion. From our results in Figure 3(a), convection is likely triggered once near-surface omega becomes negative, which is about 30 minutes prior to time=0. Prior to time=0, dCAPELSFT is nearly constant and thus cannot be used to detect initiation of convection. We did, as the reviewer suggested, investigate the relationship between d(dCAPELSFT)/dt and the precipitation. In Additional Figure 3 d(dCAPELSFT)/dt is in red and convective precipitation is in black, both computed using the spCAM5 fields from Figure 3(a) in the manuscript. From Additional Figure 3, d(dCAPELSFT)/dt becomes positive about 20 minutes prior to time=0, and reaches its maximum slightly prior to the precipitation maximum. We have not investigated these findings further, and will leave them for future study. One possibility is that the d(dCAPELSFT)/dt trend might be linked to the trend in the near-surface omega in Figure 3(a).

***As shown in Song and Zhang (2018), the dCAPE-type triggers are significantly scale dependent. In the higher-resolution models, it doesn't work very well compared to the coarser model resolution, since the relationship between dCAPE and convective precipitation becomes worse when the resolution is increased. Here, the spCAM5 is 4 km and CanAM4 is about 300 km. From figure 2, it seems that the relationship between convective precipitation and dCAPELSFT is much closer in CanAM4. Could you calculate the correlation and make some discussion about this issue. Reference: Song, F. and G. Zhang, 2018: Full Access Understanding and Improving the Scale Dependence of Trigger Functions for Convective Parameterization Using Cloud-Resolving Model Data, Journal of Climate, 7385-7399.***

We used all 32 CRM columns to compute an average spCAM5 convective precipitation and compare this "low-resolution" precipitation to CanAM4.3 convective precipitation. We have not investigated the dependence of spCAM5 precipitation to the number of CRM columns used to compute an average convective precipitation, because that is out of the scope of this paper. We did, however, compute the linear Pearson correlation

coefficient between dCAPELSFT and the convective precipitation and we found that the correlation is higher in CanAM4.3 (0.68) than in spCAM5 (0..44).

*Finally, in the spCAM5, dCAPELSFT cannot be regarded as pure large-scale forcing, since it is calculated based on 4km dataset (also see the discussion in Song and Zhang 2018). So how the convection happens in this model should be investigated further, since it provide more accurate description of convection. That will provide more information to the community.*

DCAPELSFT was computed using the large-scale T and Q fields and the large-scale T and Q spCAM5 tendencies. It is true that the large-scale tendencies include small-scale tendencies due to various processes that occur within the CRM (4-km) columns, but we should clarify that our goal was to try understand the overall effect of these 4-km small-scale tendencies on the convective precipitation in spCAM5. The overall effect of these small scale tendencies are therefore directly comparable with the overall tendencies generated within the Zhang-McFarlane convection scheme employed in CanAM4.3. The manuscript shows the differences between the overall precipitation and large-scale forcing fields between the two models.

We added the following text to Section 4.3 in the manuscript:

"Therefore, a transition from a large-scale subsidence to large-scale ascent may be important in triggering convection. A near-surface omega tendency has been previously used as a trigger in the Donner convection scheme (Donner 1993; Donner et al. 2001; Wilcox and Donner 2007) in a version of the Geophysical Fluid Dynamic Laboratory (GFDL) Atmospheric model, version 3 (AM3) GCM. In their model, convection is triggered when near-surface omega becomes positive and exceeds a specified value and convective inhibition is less than 100 J kg-1.".

[Figure]

*Additional Figure 2*

[Figure]

*Additional Figure 3*

---

## Author Comment (AC2) · 16 Jan 2019

Thank you for your constructive comments, which were useful to improve our paper. Please see our responses below. The comments are in **bold** *italics* while the responses are in normal type.

***I encourage the authors to provide a deeper discussion and interpretation of the results. For example, the lack of relationship between convective precipitation and near surface vertical velocity (Figure 2b, 3b) and the mismatch in timing with CAPE/ CIN in CanAM4.3 relative to spCAM5 are particularly interesting findings.***

We added the following text in Section 4.4 of the manuscript:

"Therefore, a transition from a large-scale subsidence to large-scale ascent may be important in triggering convection. A near-surface omega tendency has been previously used as a trigger in the Donner convection scheme (Donner 1993; Wilcox and Donner 2007) in a version of the Geophysical Fluid Dynamic Laboratory (GFDL) Atmospheric model, version 3 (AM3) GCM. In their model, convection is triggered when near-surface omega becomes positive and exceeds a specified value and convective inhibition is less than 100 J kg-1".

"Convective precipitation in CanAM4.3 does not seem to correlate well with CIN (Fig. 3b) and this is likely because CIN is not independently included in the ZM closure in CanAM4.3. Therefore, any discussion of CIN and linkage to CanAM4.3 precipitation is out of the scope of this paper. We should point out thought that, CIN is tightly coupled with precipitation over mid-latitude summertime continent but not with precipitation over oceans (Myoung and Nielsen-Gammon, 2010)."

***How are the deficiencies in the parameterizations used in CanAM4.3 (i.e., CAPE based closure), which have been identified here, different from what is already known and published?***

In the introduction we briefly mentioned that some commonly used convective scheme in GCMs employ triggers and closures based on convective available potential energy (CAPE) or CAPE generation while other closures are based on net column moisture convergence. Other convective schemes, for instance the Donner convective scheme, use grid-scale upward motion in the lower troposphere as trigger function. Although, the Zhang-McFarlane (ZM) convection scheme is very popular and has been modified and improved over time, as described for example in Zhang and Mu, 2005a, the ZM scheme still has deficiencies, such as, generates too frequent too light precipitation and underestimates the frequency of extreme events. However, various models employ various version of the ZM scheme and our goal is not to modify the ZM scheme employed in CanAM4.3, but to compare the precipitation generated within the ZM scheme to precipitation generated within a cloud-resolving model under similar large-scale forcings.

***And how can new information from the results presented here be applied to further improve models beyond what has already been implemented?***

One new result is that precipitation generated within a cloud-resolving model depends on both CAPE generation and near-surface omega, two commonly used variables in the trigger and closure functions of most popular convective parameterization schemes, while convective precipitation is a function of CAPE only in a CAPE based closure model (CanAM4.3). Another new result is that, the cause-consequence analysis (Figure 3) show that variations in omega precede variations in convective precipitation while variations in CAPE generation trail variations in convective precipitation in spCAM5. Based on these results we suggested that near-surface omega might qualify as better trigger and may be used together with dCAPELSFT in the closure scheme in CanAM4.3.

*The effort to calculate "convective precipitation in spCAM5" in a way that is comparable to "convective precipitation in a parameterized model" is a great idea and potentially very useful. However, it is not clear that the way it is calculated in spCAM5 here means the same thing as it does from parameterized convection in CanAM4.3. How sensitive are the results to the values of the criteria (vertical velocity and cloud water/ice)? More importantly, how well does a definition of "convective precipitation " based on CRM vertical velocity and cloud water/ice match what "convective precipitation" means in a global parameterized model? Since the comparison and analysis is contingent on this calculation, it would be useful to discuss other ways it could be defined within spCAM5 and/or expected differences with what convective precipitation means in CanAM4.3. It would also be helpful to use an independent calculation of "convective precipitation" that could be applied identically to both models, which would likely be dependent on large-scale conditions. Ultimately, to what degree do the results and comparison between the models depend on the way that convective precipitation has been defined? Likewise, how is CAPE calculated in spCAM5, is it at the CSRM or GCM scale? A comparison to CAPE calculated at the GCM scale would be most consistent with CAPE from CanAM4.3. Along these same lines, the differences in the relationship of convective precipitation and omega between spCAM5 (strong correlation) and CanAM4.3 (no correlation) may be, in part, due to differences in the definition of convective precipitation. I suggest including some analysis of relationships with "total precipitation rates" or alternative definition of "convective precipitation" in spCAM5.*

We agree with the reviewer that the definition of convective precipitation in spCAM5 will be sensitive to the values of the vertical velocity and the cloud water and ice. The method we used to define convective precipitation follows that in Suhas and Zhang (2015) and Song and Zhang (2018). Using this method, 68 % of the total precipitation in spCAM5 was convective compared to 71 % in CanAM4.3.

However, as the reviewer suggested, we further investigated the sensitivity of our results to the definition of convective precipitation. We repeated all the analyses using total precipitation instead of convective precipitation and generated Additional Figure 1, 2(a), 2(c), and 3 (below). The results in the Additional Figures are similar to those in Figure 1, 2(a), 2(c), and 3 in the manuscript. Therefore, we can say that the findings in the manuscript are not sensitive on the details of how the rainfall is partitioned in both spCAM5 and CanAM4.3.

We added the following text in Section 3.1 of the manuscript:

"The sensitivity of the results to the definition of convective precipitation from spCAM5 was evaluated by repeating the analyses using total instead of convective precipitation. The results in Figure 1, 2(a), 2(c), and 3 were found to be similar using either the total or convective precipitation from spCAM5, implying insensitivity, for this study, to the exact definition of thresholds in the method of Suhas and Zhang (2015)."

*In general, an explicit inclusion of observations for comparison would be helpful to the reader. The authors note that there is no dependence of convective precipitation with CAPE in spCAM5, which they say is consistent with observations by citing Mitovski and Folkins [2014]. It would be useful to make this calculation and include the observations in the figure for both CAPE and dCAPE. Likewise, the authors note that spCAM5's relationship between min/max CAPE and the timing of rainfall is consistent with observations by referring to Mitovski and Folkins [2014], but again I think showing the actual observations (as referenced) on the same figure would help.*

Mitovski and Folkins 2014 used 12-hour vertical profiles of temperature and specific humidity to compute CAPE. In addition, they used 3-hour TRMM 3B42 rainfall to isolate rainfall events. In this paper, however, we use sub-hourly model data to compute CAPE and investigate the relation with convective precipitation. Although the temporal resolution of the data used in Mitovski and Folkins (2014) and in this paper is different, it has been previously shown that tropical convection exhibits similar behavior on various time-scales (Mapes et al., 2006: "The mesoscale convection life cycle: Building block or prototype for large-scale tropical waves?"). The similarity in the observed and simulated (spCAM5) CAPE variation, once again shows that, in absence of higher-resolution observations, spCAM5 may be useful in studying convection-large-scale environment interactions.

To make it clear that Mitovski and Folkins 2014 use 12-hour soundings, we added the following text in Section 4.4 of the manuscript:

"12-hourly"

*Minor Comments:*

*Why not evaluate the ZM scheme as implemented in the conventional CAM5 to have more consistency with spCAM5? Many other aspects of the model are different between CanAM4.3 and spCAM5, beyond just the representation of convection, which makes the comparison somewhat unconstrained. I suggest including results from CAM5 as well as CanAM4.3 and spCAM5. Since only 3 months of simulation time is being assessed here and the initial setup of CAM5 would be the same as sp-CAM5, this should not add a significant amount of work.*

We agree that it would be interesting and useful to perform the analysis using CAM5 simulations that are configured the same as spCAM5. However, this is a non-trivial amount of work due since the spCAM5 data we used was archived from previous simulations and we no longer have access to the personnel and computer accounts. To perform the CAM5 would require significant effort to set as it would require setting up the model on a new computer system with all of the associated effort to verify it is

implemented correctly. Repeating the analysis with CAM5, and potentially other model, would be something that could performed in future studies.

*I am confused about the vertical resolution used in spCAM5. Typically, the vertical resolution is 30 levels in the global grid and 28 levels in the CSRM (coinciding with the lowest 28 levels). Here the authors state that there are 66 levels CAM5, which would imply 38 levels above the CSRM rather than the typical 2 levels. Have previous studies used this configuration? Have you evaluated the differences between using 30 and 66 levels Additionally, the Khairoutdinov and Randall (2001 and 2003) references are fairly old and refer to the implementation of super-parameterization in older versions of CAM. I recommend the authors cite more recent papers describing the implementation in CAM5, such as Wang et al., 2011 ([https://www.geosci-model-dev.net/4/137/2011/gmd-4-137-2011.pdf](https://www.geosci-model-dev.net/4/137/2011/gmd-4-137-2011.pdf)).*

We stated all CAM5 and CanAM4.3 levels in the atmosphere. As the reviewer suggester, we updated the manuscript and include the lower atmosphere levels only, as well as, we cite Wang et al., 2011. We substituted the following text in Section 2:

"66 vertical levels from the surface to 5.1 x 10e-6 hPa "

With:

"30 vertical levels from the surface to 3.6 hPa"

*Since spCAM5 is used instead of spCAM4, it includes aerosol processes and two-moment microphysics, so it might be helpful to describe these components of the model (MAM3 aerosol and Morrison microphysics) and compare them with the same processes in CanAM4. The representation of aerosol and cloud microphysics are likely to influence precipitation as well.*

We agree that it is possible that the aerosol and cloud microphysics formulation could influence the precipitation but our hypothesis is that the main control in the Tropical Western Pacific is rainfall from the deep convective scheme. In the paper we note that roughly 70% of the rain is convective and it seems that the stratiform rain has little effect on the results (performing the analysis using the total rain or the convective rain give similar results). We leave it to the interested reader to refer to the references for details about the aerosol and cloud microphysics parameterizations.

*For the relationship between vertical velocity and convective precipitation in CanAM4.3 (Figure 1a), the authors conclude that "the results are not considered robust due to the few samples". Why not use more years for the CanAM4.3 results? CanAM4.3 is relatively cheap to run, so it is unnecessary for the authors to limit their analysis to such a short period. I recommend using more data, at least for CanAM4.3, to produce more robust results*

We found this suggestion very useful and we therefore  perform another five CanAM4.3 ensemble simulations for the period of study. The ensemble was generated by changing the random number seed on 1 January 1997.  We incorporated the data from these simulations into our analysis. Figures 1, 2, and 3, are now based on 5 CanAM4.3 ensemble runs.

We added the following text in Section 2 of the manuscript:
"For CanAM4.3, a six member ensemble was generated by uniquely adjusting the seed for the random number generator on 1 January 1997.  This was done to improve the statistical representation of the results from this model as data from all ensemble members were used in the analysis'.

[Figure]

*Additional Figure 1*

[Figure]

[Figure]

*Additional Figure 2(a) and 2(c)*

[Figure]

*Additional Figure 3*

---

## Author Response (AR2)

Thank you for your thoughtful and constructive comments. Comments are in **bold** *italics* while our responses are in normal text.

***This paper is well written and the results will be of interest to the model development community. Particularly, the use a model like spCAM5, which better resolves convective processes, to learn how we might improve convective parameterizations. Overall, the authors have responded well to many of the reviewer comments, but have not necessarily reflected the responses as edits to manuscript. Additionally, the manuscript would likely have a larger impact if some of the remaining comments were addressed directly. I outlined some remaining issues below that I believe would make the work more impactful and leave it to the editor to decide if these are within the scope of this work.***

***I think some of the responses to both reviewers' comments add important insight and could be better incorporated into the main manuscript. In particular, clarifying the how the results of this work that are new and discussing how they might be applied to guide future model development would be useful. I also suggest incorporating the "additional figures" into the main manuscript or as supplemental material, which will likely be of interest to many readers.***

The additional figures from our previous responses are now included in the supplemental as Figs. S1, S2, and S3. Also included in the supplemental are the dCAPE$_{LSFT}$ tendency as Fig. S4. As outline below some of previous responses to reviewers' comments have been incorporated into manuscript main text.

The following text was added to Section 4.3:

"Repeating Fig. 3, using dCAPE$_{LSFT}$ and omega at levels ranging from 992 hPa to 232 hPa we found that, greater rainfall rates in spCAM5 is associated with more negative (stronger ascent) omega at 992 hPa and less negative omega at 232 hPa (not shown). Since omega was computed using Eq.4, a negative omega at pressure 992 hPa is approximately equal to the net column mass divergence above that pressure level.. Therefore, greater rainfall rates in spCAM5 are associated with strong low-level ascent or strong net column mass divergence and larger dCAPE$_{LSFT}$."

The following text was added to Section 4.4:

"In addition to examining the time evolution of dCAPE$_{LSFT}$ over convective events we compared its tendency (time change) and convective precipitation in spCAM5 (Fig. S4). We found that the tendency in dCAPE$_{LSFT}$ (Fig. S4) becomes positive about 20 minutes prior to the start of a convective event ($t$=0) reaching a maximum slightly prior the precipitation maximum. While a thorough examine of this is beyond the scope of this paper we hypothesize that the trend in dCAPE$_{LSFT}$ could be associated with the trend in near surface omega. I. e.. an increasing trend in the large-scale ascent contributes to increasing CAPE and positive trend in large-scale CAPE generation. "

The following text was added to Section 5:

"Convective precipitation generated within spCAM5 is found to depend on both CAPE generation rate and near surface vertical velocity, two fields commonly used in trigger and closure functions of

convective parameterization schemes.  In CanAM4.3, which is representative of a CAPE-based closure scheme. convective precipitation is found to be a function of CAPE only (or CAPE generation).

For example, our results suggest that near-surface omega might provide a better trigger in combination with CAPE generation in the closure scheme used within CanAM4.3.”

*The additional figures using total instead of convective precipitation do not completely alleviate the issues about how "convective precipitation" is defined. It is not enough to show that the percent defined as convective is similar in spCAM5 as in CanAM4.3, or that the total and convective precipitation in spCAM5 shows a similar result. These do not necessarily indicate the offline calculation of "convective precipitation" means the same thing it does in a parameterized model.*

The following text was added to Section 3.1 to better explain the source of the method in spCAM5:

“This definition is used in the studies of Suhas and Zhang (2015) and Song and Zhang (2018) which follows the study of Xu and Randall (2001) who set the thresholds based on examination of convective updraft and downdraft statistics in cloud-resolving model simulations of tropical and midlatitude convection.”

and the following reference to References:

Xu, K.-M., and D. A. Randall, 2001: Updraft and downdraft statistics of simulated tropical and midlatitude cumulus convection. *J. Atmos. Sci.*, **58**, 1630–1649, https://doi.org/10.1175/1520-0469(2001)058<1630:UADSOS>2.0.CO;2.

*Would it be possible to calculate the "convective precipitation" from "total precipitation" in CanAM4.3 using the same method used for spCAM5, then compare to what CanAM4.3 internally defines as “convective precipitation”? If an offline calculation of "convective precipitation" gives the same result as the modeled "convective precipitation" in CanAM4.3, it would support its use with spCAM5.*

We do not feel that this is possible.  The method to define convective precipitation in spCAM5 is meant to be applied to individual CSRM columns so applying it to a GCM grid box does not seem reasonable.

*Additionally, there are differences for CanAM4.3 in the relationship between “mean peak precipitation” and “event length” when "total" is used instead of "convective" precipitation (Figure 1d vs. Additional Figure 1d). In particular, the total precipitation shows an increase with event length. Does this imply that large-scale precipitation plays an important role for longer events?*

The following text was added to Section 4.2:

“To examine sensitivity to the definition of convective precipitation (Sec. 3.1) we repeated our analysis using total instead of convective precipitation (Fig. S1).  The results are similar except for the length of convective events and peak precipitation (Fig. 1d and Fig. S1d).  The difference is due mainly to CanAM4.3 frequently producing light non-convective precipitation.  This affected the event length

while peak rainfall was more influenced by convective precipitation suggesting that non-convective precipitation is more important for long lasting events."

***For additional Figure 2 and 3, it would a be useful to show CanAM4.3 total precipitation in order to assess consistency with the convective precipitation results alongside spCAM5.***

We now include CanAM4.3 total precipitation in Supplemental Figure 2 and 3.

We added the following text to Section 3.1:

(Figs S1, S2, and S3)

This is mainly because in the regions being studied the precipitation was found to be mainly convective (> 70 %).

***It makes sense that spCAM5 could be a proxy "in the absence of high spatial resolution and sub-hourly observations". However, it would be useful to show that spCAM5 can capture the relationships seen in nature at the space-time resolutions that observations are available. It would make a much stronger case to include some comparison to these relationships, even at lower temporal resolution, to demonstrate that how well spCAM5 can match observations.***

We agree that this is a good idea. Unfortunately we do not have high frequency observations of convective precipitation to compare with spCAM5. Instead we compared the frequency density of the total precipitation, derived from 3-hour means, simulated by spCAM5 against that from TRMM 3B42v7. We found similar behavior i.e. GCMs tend to underestimate the frequency of the precipitation rates above 1 mm/h and overestimate light precipitation rates.

We included TRMM 3B42v7 3-hourly precipitation estimates in a separate Figure 2b. We also added the following text to Section 2:

"While observations of sub-hourly precipitation are not available in the region used in this study, 3-hour observations of total precipitation is available from TRMM (Kummerow et al, 1998). To compare with the 3-hour TRMM data, 3B42v7 (TRMM, 2011), the spCAM5 and CanAM4.3 precipitation rates were averaged to 3-hour means.

The frequency distribution for total precipitation (Fig. 2b) shows that spCAM5 is more similar to TRMM than CanAM4.3 and CAM5.1 both of which are in turn more similar to each other than spCAM5 or TRMM. The similarity between CanAM4.3 and CAM5.1 and their difference relative to spCAM5 also holds for convective precipitation (Fig. 2a). These results suggest, at least for 3-hour means, the exact definition of convective rainfall is less important than differences between CanAM4.3 and spCAM5."

We added the reference:

Kummerow, C., Barnes, W., Kozu, T., Shiue, J., and Simpson, J.: The Tropical Rainfall Measuring Mission (TRMM) sensor package, J. Atmos. Oceanic Technol., 15, 809–817, doi:10.1175/1520-0426(1998)015<0809:TTRMMT>2.0.CO;2, 1998.

*I think the paper would be stronger if CAM5 results were included in some way to address the unconstrained differences between spCAM5 and CanAM4.3. If running CAM5 is too difficult, I would encourage the authors to look for a similar simulation available in archives (ESGF or NCAR) of CAM5 data to enable at least a qualitative comparison. In the absence of including CAM5 results, it is important to clarify in more detail what all the relevant differences between spCAM5 and CanAM4.3 might be. I do not think it is sufficient to simply provide a reference, a summary of the differences, including microphysics and aerosol, would be helpful.*

As noted previously we were unable to run CAM5. Searching the publicly available archive of simulations performed by NCAR we were able to find CAM5.1 AMIP simulations which overlapped our period and region of interest. Given the specialized nature of the model output used in this manuscript we were only able to find 3-hour means of total and convective precipitation. This significantly limited our use of CAM5.1 model output but we did include it in our new Figure 2. This shows that even with all of the differences between CanAM4.3 and CAM5.1 they are more similar to each other than to spCAM5.

While not conclusive or quantitative, this suggests to us that the use of subgrid-scale parameterizations rather than CSRMs are a larger effect than the unconstrained differences between CanAM4.3 and spCAM5. It also suggests to use that the definition of convective precipitation seems to be smaller than the differences between CanAM4.3, CAM5.1 and spCAM5.

We added the following text to Section 4.2:

[revised manuscript text omitted]